# Metabolic imbalance limits fermentation in microbes engineered for high-titer ethanol production

Bishal Dev Sharma,[1,2] Eashant Thusoo,[2,3,4] David M. Stevenson,[2,3,4] Daniel Amador-Noguez,[2,3,4] Lee R. Lynd,[1,2,5] Daniel G. Olson[1,2]

**ABSTRACT** Microbial strains engineered for high-titer ethanol production often stop fermenting while substantial substrate remains, limiting industrial performance. We investigated this limitation in engineered strains of *Escherichia coli* and *Thermoanaerobacterium saccharolyticum* and the native ethanologen *Zymomonas mobilis*. By combining high-titer fermentations with intracellular metabolomics, we are able to see how intracellular metabolite concentrations change as product formation stops. We then used max-min driving force (MDF) thermodynamic analysis to understand how these changes in intracellular metabolite levels can limit flux and to identify key enzymes that might be responsible for these limitations. In engineered strains, cessation of ethanol production coincided with strong pyruvate accumulation and MDF values near or below zero at the pyruvate kinase step, implying that the pyruvate consuming enzyme(s) (pyruvate decarboxylase for *E. coli* and pyruvate ferredoxin oxidoreductase, or associated electron transfer enzymes for *T. saccharolyticum*) might limit flux. By contrast, *Z. mobilis* maintained positive driving forces without pyruvate buildup, suggesting that its titer is limited by processes outside central carbon metabolism, such as substrate uptake. These results establish a generalizable framework linking metabolite concentrations to pathway thermodynamics and demonstrate how thermodynamic analysis can diagnose where metabolic constraints emerge during high-titer fermentation.

**IMPORTANCE** High-titer fermentation is essential for economically viable biofuel production, yet even extensively-engineered microbes frequently stop producing ethanol before the substrate is exhausted. Furthermore, the causes of titer limitations are often poorly understood. A particular challenge is identifying the location of titer limitations in multi-enzyme pathways. Here, we show that MDF analysis can assist in the interpretation of metabolomic data. These findings provide a systems-level explanation for "stuck" fermentations in bacteria and identify thermodynamic driving force as a quantitative diagnostic metric that reveals where biological design targets emerge for metabolic engineering of ethanol and other bioproducts.

**KEYWORDS** biofuel, *Escherichia coli*, max-min driving force, metabolomics, *Thermoanaerobacterium saccharolyticum*, thermodynamics, *Zymomonas mobilis*

Fossil fuels have been a central energy source for global industrialization and transportation for centuries. However, growing concerns about their limited supply and their significant contribution to climate change have highlighted the need for a transition to sustainable fuel sources. While electric vehicles are making progress, dense liquid fuels remain essential, particularly for long-haul trucking, ocean shipping, and aviation (1, 2). Among the options for renewable liquid fuels that can be produced from cellulose, ethanol remains one of the most promising due to its relatively low toxicity to microorganisms, its ability to be produced at high yield and titer, and its ease of separation by distillation (2).

**Peer Reviewer** Uldis Kalnenieks, University of Latvia, Riga, Latvia

Address correspondence to Daniel G. Olson, daniel.g.olson@dartmouth.edu.

The authors declare no conflict of interest.

See the funding table on p. 21.

For a standalone production facility, ethanol titers need to be in the range of 40–50 g/L to reduce costs associated with fermenter size and distillation (2, 3). Several efforts have been made to engineer microbes capable of producing ethanol at industrially relevant titers. To date, only three microbes—*E. coli*, *T. saccharolyticum*, and *Corynebacterium glutamicum*—which do not naturally produce ethanol as a major product, have been successfully engineered to achieve ethanol titers higher than 50 g/L (4–6). The engineering of *E. coli* involved deletion of genes responsible for the production of other fermentation products (i.e., acetate and lactate), along with the deletion of the native alcohol dehydrogenase gene and the introduction of genes for pyruvate decarboxylase and alcohol dehydrogenase from *Zymomonas mobilis* (6). The same pathway was introduced into *C. glutamicum* (7). Subsequently, overexpression of glycolytic genes and optimization of fermentation conditions resulted in ethanol production at a titer of 106 g/L (5).

Similarly, for *T. saccharolyticum*, engineering involved the deletion of genes for competing products such as lactate and acetate (4). Unlike these engineered strains, *Z. mobilis* is a natural ethanologen and the most productive bacterial ethanol producer, capable of achieving ethanol titers up to 127 g/L in batch fermentation (8, 9) while retaining high yield and productivity, making it an important benchmark for microbial ethanologen design (9–11). However, despite its efficiency, *Z. mobilis* has limited substrate range (i.e., it only ferments glucose, fructose, and sucrose) and low thermotolerance, which constrains its use for lignocellulosic biofuel production. Nevertheless, its physiology and metabolic characteristics continue to provide valuable insights for the study and engineering of ethanologenic microbes.

The ability to transfer phenotypes from one organism to another organism is a cornerstone of modern biotechnology. Ethanol production is one of the first phenotypes for which this was demonstrated (12). Although it is relatively trivial to engineer an organism to produce small amounts of ethanol, it is much more difficult to transfer the industrially relevant phenotype of ethanol production at the high yield, titer, and rate that is present in native producers such as *Z. mobilis*.

In this work, we identified fermentation conditions where ethanol production stops despite the presence of residual substrate for *E. coli*, *T. saccharolyticum*, and *Z. mobilis*. Using these fermentation conditions, we then quantified changes in glycolytic metabolite concentrations to observe the dynamic changes to metabolism as growth and fermentation stop. Finally, we used the max-min driving force (MDF) framework (13) to understand how these changes in metabolite concentrations constrain flux through key metabolic pathways including glycolysis and fermentation.

By studying why ethanol production stops in engineered ethanol producers, such as *E. coli* and *T. saccharolyticum*, and comparing them with native producers such as *Z. mobilis*, we characterize the extent to which engineered strains recapitulate the industrially relevant ethanol production phenotype, as well as the mechanistic basis for existing limitations to product titer. We expect that the improved understanding of high-titer ethanol production developed in this work will build a strong foundation for efforts to transfer this phenotype to other organisms (e.g., *Clostridium thermocellum*) or even whole microbial communities, and will also inform strategies for producing other bioproducts at high yield and titer.

## MATERIALS AND METHODS

### Strains used in this work

Strains used in this work are shown in Table 1.

### Enzymes and metabolites

Metabolites discussed in this work are referred to as follows: 2-phosphoglycerate, 2pg; 3-phosphoglycerate, 3pg; 6-phosphogluconolactone, 6pgln; 6-phosphogluconate,

**TABLE 1** Strains used in this study[a]

| Organism | Strain ID | Genotype | Reference | Remarks |
|---|---|---|---|---|
| *E. coli* | RL3019 or LL1777 | WT Δ*frdA*, Δ*ldhA*, Δ*ackA*, Δ*adhE* with pB1 plasmid (*Zymomonas mobilis pdc, adhB*) | (14) | Engineered mesophilic ethanologen based on widely used *E. coli* K-12 MG1655 strain |
| *T. saccharolyticum* | M1442 or LL1049 | WT Δ(*pta-ack*) Δ*ldh* Δ*or795::metE-ure* Δ*eps* | (4) | Engineered thermophilic ethanologen. Ethanol titers up to 70 g/L reported |
| *Z. mobilis* | ZM4 or LL1719 | WT | ATCC 31821 | Natural ethanologen used as benchmark control |

[a]Note: LL strain IDs are internal designations from the Lynd and Olson lab strain collection at Dartmouth.

6pgn; acetaldehyde, acald; acetyl coenzyme A, accoa; adenosine diphosphate, adp; adenosine triphosphate, atp; bisphosphoglycerate, bpg; cellobiose, cb; coenzyme A, coa; dihydroxyacetone phosphate, dhap; ethanol, etoh; fructose 6-phosphate, f6p; fructose 1,6-bisphosphate, fbp; oxidized ferredoxin, fdox; reduced ferredoxin, fdred; glycerol-3-phosphate, g3p; glucose 6-phosphate, g6p; glucose, glc; 2-keto-3-deoxy-6-phosphogluconate, kdpg; oxidized nicotinamide adenine dinucleotide, nad; reduced nicotinamide adenine dinucleotide, nadh; oxidized nicotinamide adenine dinucleotide phosphate, nadp; reduced nicotinamide adenine dinucleotide phosphate, nadph; phosphoenolpyruvate, pep; pyruvate, pyr.

Enzymes discussed in this work are referred to as follows: alcohol dehydrogenase reaction (NAD dependent), ADH; alcohol dehydrogenase reaction (NADP dependent), ADHP; aldehyde dehydrogenase reaction, ALDH; cellobiose phosphorylase reaction, CBP; 2-keto-3-deoxy-6-phosphogluconate aldolase, EDA; 6-phosphogluconate dehydrogenase, EDD; enolase reaction, ENO; fructose-1,6-bisphosphate aldolase reaction, FBA; ferredoxin:nadp oxidoreductase reaction, FNORP; glucose 6-phosphate dehydrogenase reaction, G6PDH; glyceraldehyde-3-phosphate dehydrogenase reaction, GAP; glucokinase reaction, GLK; phosphoglycerate mutase reaction, GPM; pyruvate decarboxylase reaction, PDC; pyruvate:ferredoxin oxidoreductase reaction, PFOR; phosphoglucose isomerase reaction, PGI; phosphoglycerate kinase reaction, PGK; 6-phosphogluconolactonase, PGL; phosphoglucomutase reaction, PGM; phosphofructokinase reaction, PFK; pyruvate kinase reaction, PYK; triosephosphate isomerase reaction, TPI.

## Media and growth conditions

All reagents used in this study were of molecular grade and obtained from Sigma Aldrich or Fisher Scientific, unless otherwise noted.

*E. coli* cultures were grown anaerobically at 37°C in either lysogeny broth (LB) medium (BP1427-500, Fisher Scientific) or M9 minimal medium modified to contain calcium and iron (15) with addition of betaine. M9 minimal medium contained a final concentration of 120 g/L glucose, 6.7 g/L $Na_2HPO_4$, 3 g/L $KH_2PO_4$, 0.5 g/L NaCl, 1 g/L $NH_4Cl$, 0.246 g/L $MgSO_4$, 0.0011 g/L $CaCl_2$, 0.0002 g/L $FeSO_4 \cdot 7H_2O$, 0.1536 g/L betaine hydrochloride, and 50 µg/mL spectinomycin. The medium components were prepared as separate solutions. M9 salts, 5-fold concentrated, contained $Na_2HPO_4 \cdot 7H_2O$, $KH_2PO_4$, NaCl, and $NH_4Cl$ (M6030-1KG, Sigma-Aldrich). $MgSO_4$, $CaCl_2$, $FeSO_4 \cdot 7H_2O$, and betaine hydrochloride solutions were prepared at 500-fold, 100-fold, 100-fold, and 1,000-fold concentrations, respectively. All the solutions were filter-sterilized and stored under nitrogen atmosphere. Spectinomycin (2,000-fold concentrated) was obtained from Sigma-Aldrich (catalog no. S0692-1ML).

*T. saccharolyticum* cultures were grown anaerobically at 51°C in either TSC6 medium or MTC-7 medium with or without yeast extract. The TSC6-rich medium contained a final concentration of 90 g/L cellobiose and 60 g/L maltodextrin, 10 g/L calcium carbonate, 8.5 g/L yeast extract, 0.5 g/L trisodium citrate·2H2O, 2.0 g/L $KH_2PO_4$, 2.0 g/L $MgSO_4 \cdot 7H_2O$, 5 g/L urea, 0.2 g/L $CaCl_2 \cdot 2H2O$, 0.2 g/L $FeSO_4 \cdot 7H_2O$, 0.12 g/L methionine, and 0.5 g/L L-cysteine HCl, as described previously (4, 16).

The MTC-7 chemically defined medium contained a final concentration of 140 g/L cellobiose, 9.3 g/L MOPS (morpholinepropanesulfonic acid) sodium salt, 2 g/L potassium citrate monohydrate, 1.3 g/L citric acid monohydrate, 1 g/L $Na_2SO_4$, 1 g/L $KH_2PO_4$, 2.5 g/L $NaHCO_3$, 2 g/L urea, 1 g/L $MgCl_2 \cdot 6H_2O$, 0.2 g/L $CaCl_2 \cdot 2H_2O$, 0.1 g/L $FeCl_2 \cdot 4H_2O$, 1 g/L L-cysteine HCl monohydrate, 0.02 g/L pyridoxamine HCl, 0.004 g/L p-aminobenzoic acid, 0.002 g/L D-biotin, 0.002 g/L vitamin B12, 0.004 g/L thiamine, 0.0005 g/L $MnCl_2 \cdot 4H_2O$, 0.0005 g/L $CoCl_2 \cdot 6H_2O$, 0.0002 g/L $ZnCl_2$, 0.0001 g/L $CuCl_2 \cdot 2H_2O$, 0.0001 g/L $H_3BO_3$, 0.0001 g/L $Na_2MoO_4 \cdot 2H_2O$, and 0.0001 g/L $NiCl_2 \cdot 6H_2O$, as described previously (15, 17). The medium components were prepared as separate solutions (A–F). Solution A, concentrated 1.2-fold, contained cellobiose and MOPS sodium salt. Solution B, concentrated 25-fold, contained potassium citrate monohydrate, citric acid monohydrate, $Na_2SO_4$, $KH_2PO_4$, and $NaHCO_3$. Solution C, concentrated 25-fold, contained urea. Solution D, concentrated 50-fold, contained $MgCl_2 \cdot 6H_2O$, $CaCl_2 \cdot 2H_2O$, $FeCl_2 \cdot 4H_2O$, L-cysteine HCl monohydrate, and trace minerals (solution F). Solutions E, concentrated 50-fold, contained pyridoxamine HCl, p-aminobenzoic acid, D-biotin, vitamin B12, and thiamine. Solution F, concentrated 1,000-fold, contained $MnCl_2 \cdot 4H_2O$, $CoCl_2 \cdot 6H_2O$, $ZnCl_2$, $CuCl_2 \cdot 2H_2O$, $H_3BO_3$, $Na_2MoO_4 \cdot 2H_2O$, and $NiCl_2 \cdot 6H_2O$. For medium without active pH control, the initial pH was maintained at 6.5 using 10% (wt/vol) $H_2SO_4$. For medium with yeast extract, 8.5 g/L yeast extract was added to MTC-7. Note: 140 g/L is near the upper limit of solubility for cellobiose but can solubilize at concentrations as high as 200 g/L when autoclaved.

*Z. mobilis* cultures were grown anaerobically at 37°C in *Zymomonas mobilis* minimal (ZMM-2) medium supplemented with methionine and lysine (18). ZMM medium contained a final concentration of 20–160 g/L glucose, 1 g/L $KH_2PO_4$, 1 g/L $K_2HPO_4$, 0.5 g/L NaCl, 1 g/L $(NH_4)_2SO_4$, 0.2 g/L $MgSO_4 \cdot 7H_2O$, 0.025 g/L $NaMoO_4 \cdot 2H_2O$, 0.0025 g/L $FeSO_4 \cdot 7H_2O$, and 0.02 g/L calcium pantothenate. ZMM-2 had an additional 2 g/L methionine and 2 g/L lysine. The medium components were prepared as separate solutions. ZMM salts, concentrated 10-fold, contained $KH_2PO_4$, $K_2HPO_4$, NaCl, and $(NH_4)_2SO_4$. All other components, except glucose, were concentrated 1,000-fold. Concentrated glucose solution of 400 g/L was prepared by filter sterilization. For the growth medium without active pH control, the initial pH of the medium was maintained at 6.0 using 10% (wt/vol) $H_2SO_4$. The medium remained turbid until adjusted to pH 6.0, due to the limited solubility of lysine and methionine at higher pH. All media were filter-sterilized after combining the component solutions.

## pH-controlled bioreactor fermentation conditions

*T. saccharolyticum* cultures were grown anaerobically at 51°C in 400 mL MTC-7 medium with 140 g/L cellobiose as substrate, with pH controlled at 6.0 ± 0.05. For a 400 mL working volume with pH control in a bioreactor, 300 mL of solution A was prepared excluding MOPS sodium salt and autoclaved (note: although the solubility limit of cellobiose is reported to be ~120 g/L, autoclaving increases solubility and allows preparation of cellobiose solutions at concentrations up to 200 g/L). The bioreactor was placed in an anaerobic chamber for 12 to 16 h, with the gas inlet/outlet tube left open for purging. Then, 16 mL of solution B, 8 mL of solution C, 8 mL of solution D, and 16 mL of solution E were added through a 0.22 µm filter (catalog no. 430,517; Corning). The final volume was adjusted to 400 mL using autoclaved distilled water stored in a nitrogen atmosphere.

*E. coli* cultures were grown anaerobically at 37°C in 400 mL of M9 minimal medium with 120 g/L glucose as substrate, with pH controlled at 6.5 ± 0.05. For a 400 mL working volume with pH control in a bioreactor, the bioreactor was filled with 160 mL water and autoclaved. The bioreactor was placed in an anaerobic chamber for 12 to 16 h, with the gas inlet/outlet tube left open for purging. Then, M9 minimal medium was prepared in 2-fold concentration and stored in the anaerobic chamber. After purging, 200 mL of 2-fold concentrated M9 minimal medium was added through a 0.22 µm syringe filter.

The final volume was adjusted to 400 mL using autoclaved distilled water stored under a nitrogen atmosphere.

*Z. mobilis* cultures were grown anaerobically at 37°C in 400 mL ZMM-2 medium with 160 g/L glucose as substrate and 10 g/L initial ethanol. The pH was controlled at 6.0 ± 0.05. For a 400 mL working volume with pH control in a bioreactor, the bioreactor was filled with 160 mL water and autoclaved. The bioreactor was placed in an anaerobic chamber for 12 to 16 h, with the gas inlet/outlet tube left open for purging. Then, ZMM-2 medium was prepared in 2-fold concentration and stored in the anaerobic chamber. After purging, 200 mL of 2-fold concentrated ZMM-2 medium was added through a 0.22 µm syringe filter. The final volume was adjusted to 400 mL using autoclaved distilled water stored in a nitrogen atmosphere.

All the bioreactor fermentations were performed in a Coy (Ann Arbor, MI) anaerobic chamber with a gas phase of 85% (vol/vol) $N_2$, 10% (vol/vol) $CO_2$, and 5% (vol/vol) $H_2$. The pH was maintained using a Mettler-Toledo (Columbus, OH, USA) pH probe (Catalog No. 405-DPAS-SC-K8S), and 4 N KOH was added as needed to control the pH. The bioreactor setup is shown in Fig. S1.

## Metabolites extraction and quantification

During fermentation, the samples were taken at different time points. To quantify the fermentation products or extracellular metabolites (acetate, ethanol, formate, lactate, pyruvate), the cell culture was centrifuged, and the supernatant was processed for HPLC analysis. The metabolites were quantified using HPLC (LC-2030, Shimadzu) with refractive index (RI) and UV detection using an Aminex HPX-87H column (Bio-Rad, Hercules, CA). The column was maintained at 60°C. The mobile phase (5 mM sulfuric acid) flow rate was 0.6 mL/minute [17, 19]. The detailed protocol for HPLC method and quantification is available at protocols.io [20].

To quantify glycolysis or intracellular metabolites, the cell samples were prepared using a previously published protocol optimized for high substrate fermentations [15]. Metabolite extraction was performed in the anaerobic chamber. The metabolite extract was prepared by filtering the cell culture using vacuum filtration followed by placing the filter cell-side down for quenching in a cold metabolite extraction buffer (40% [vol/vol] acetonitrile, 40% [vol/vol] methanol, and 20% [vol/vol] water), as previously described [15, 17, 21]. For each sample, 0.5–8 mL of cells were filtered, quenched, and dried under nitrogen gas using a sample concentrator (catalog no. EW-36620-40, Cole-Parmer). After quenching and extraction, the samples were removed from the anaerobic chamber. The samples were then resuspended in LC-MS-grade water and analyzed by LC-MS. Metabolite quantification was performed using calibration curves made using known external standards. Total intracellular volume was calculated as: volume of a single cell (estimated from cell dimensions determined by microscopy) × optical density of the culture ($OD_{600}$) × cells per mL at $OD_{600}$ = 1 × vol of culture filtered, as previously described [15]. The volume of a single cell was estimated from cell dimensions using the formula $\pi \times (radius)^2 \times length$. The calculated values were approximately 0.7 µm³ (7 × 10⁻¹³ mL) for *E. coli* (BioNumber identification number [BNID] 106,614 [22]), 0.5 µm³ (5 × 10⁻¹³ mL) for *T. saccharolyticum* [23], and 4.5 µm³ (4.5 × 10⁻¹² mL) for *Z. mobilis* [24]. Cell counts at $OD_{600}$ = 1 were approximately 8.9 × 10⁸ cells/mL for *E. coli*, 4.4 × 10⁸ cells/mL for *Thermoanaerobacterium saccharolyticum*, and 2.2 × 10⁸ cells/mL for *Zymomonas mobilis*, as determined by cell counting. For metabolites without external standards, metabolites were identified by *m/z* (mass/charge) ratio and retention time. For each metabolite, peak area values were normalized by calculating z-scores across all time points. The detailed protocol for intracellular metabolite extraction, peak detection, and quantification of metabolites is available at protocols.io [25].

## Liquid chromatography-mass spectrometry (LC-MS) analysis

Metabolomics analysis was performed using a Vanquish ultra-high-performance liquid chromatography (UHPLC) system (Thermo Scientific), coupled to a hybrid

quadrupole-Orbitrap mass spectrometer (Q Exactive; Thermo Scientific) with electro-spray ionization operating in negative-ion mode, as previously outlined (26). The chromatography was conducted at 25°C using a 2.1 × 100 mm reverse-phase C18 column with a 1.7 µm particle size (Water; Acquity UHPLC BEH). Two distinct chromatography gradients were employed. The first gradient used solvent A (97:3 $H_2O$: methanol + 10 mM tributylamine) and solvent B (100% [vol/vol] methanol) with the following time schedule: 0–2.5 min, 5% (vol/vol) B; 2.5–17 min, a linear gradient from 5% (vol/vol) B to 95% (vol/vol) B; 17–19.5 min, 95% (vol/vol) B; 19.5–20 min, a linear gradient from 95% (vol/vol) B to 5% (vol/vol) B; and 20–25 min, 5% (vol/vol) B. The second gradient also used solvents A and B (100% [vol/vol] methanol), with the following: 0–2.5 min, 5% (vol/vol) B; 2.5–7.5 min, linear gradient from 5% (vol/vol) B to 20% (vol/vol) B; 7.5–13 min, 20% (vol/vol) B to 55% (vol/vol) B; 13–18.5 min, 55% (vol/vol) B to 95% (vol/vol) B; 18.5–19 min, linear gradient from 95% (vol/vol) B to 5% (vol/vol) B; and 19–25 min, 5% (vol/vol) B. The flow rate was kept at 0.2 mL/min for both gradients. Metabolites were identified based on their retention times, determined with pure standards, and their monoisotopic mass using MAVEN (27) and El-MAVEN (28) software.

## Thermodynamic analysis

Thermodynamic max-min driving force (MDF) analysis of the fermentation pathways in this study was performed using eQuilibrator (29), implemented in Python programming language (https://gitlab.com/equilibrator). For the thermodynamics analysis, the following assumptions were applied.

i. For measured metabolites, the concentration bounds were fixed to the measured values.
ii. For non-quantified metabolites, concentrations were allowed to range from 1 µM to 100 mM. Although the standard MDF framework sets the default range from 1 µM to 10 mM (13), we relaxed the upper bound because several intracellular metabolites in this study were measured at concentrations above 10 mM.

In the model, quantified metabolites were assigned identical upper and lower bounds equal to their measured concentrations, while non-quantified metabolites were allowed to vary between 1 µM and 100 mM. For metabolites measured at a concentration of zero, the lower bound was set to 1 µM. All metabolite identifiers are from the KEGG database (30–32).

## RESULTS

### Yeast extract allows high-titer ethanol production in microbes

One of our primary goals in this work was to study factors that limit fermentation, while substrate is still present. To do this, we first needed to identify suitable initial substrate concentrations. Furthermore, because yeast extract interferes with quantification of intracellular metabolites, we aimed to identify conditions where high ethanol titers could be achieved in its absence.

Previous studies on engineered *E. coli* strains have reported ethanol titers ranging from 34 to 65 g/L (6, 12, 33–36), primarily using *E. coli* B derivatives. Most of these studies used rich medium (LB) with only a few (6, 12, 33–36) using chemically defined media like NBS or AM1. While *E. coli* B strains are widely used in ethanol studies, they are not as extensively characterized as the commonly used *E. coli* K-12 strain MG1655, which has a well-documented genetic and metabolic background (37). Ethanol titers up to ~40 g/L have been reported from *E. coli* MG1655 derivatives when growth in LB medium (38), whereas the highest reported titer for MG1655 derivatives in chemically defined medium is only ~18 g/L (39). In this study, we used an MG1655 derivative (*E. coli* strain RL3019) and found that it could produce up to 43 g/L ethanol in LB medium (Table 2). In contrast, when grown in a defined M9 minimal medium, the maximum ethanol titer decreased to ~20.5 g/L (Table 2).

**TABLE 2** Ethanol production by engineered strains of *E. coli* and *T. saccharolyticum*[a]

| Microbe | Maximum ethanol titer | Media | Substrate | Yeast extract | pH | Temp (°C) | Reference(s) |
|---|---|---|---|---|---|---|---|
| *E. coli* TCS083 | 39 g/L | LB | 80 g/L glucose | 5 g/L | – | 37 | (38) |
| *E. coli* RL3019 | 43 g/L | LB | 120 g/L glucose | 5 g/L | – | 37 | This study |
| | 16 g/L | M9 minimal medium | 120 g/L glucose | 0 | – | 37 | |
| | 21 g/L | M9 minimal medium | 120 g/L glucose | 0 | 6.5 | 37 | |
| *T. saccharolyticum* M1442 | 70 g/L | TSC6 | 60 g/L cellobiose + 90 g/L maltodextrin | 8.5 g/L | – | 51 | (4) |
| | 61 g/L | MTC7 | 140 g/L cellobiose | 8.5 g/L | – | 51 | This study |
| | 34 g/L | MTC7 | 140 g/L cellobiose | 0 | – | 51 | |
| | 44 g/L | MTC7 | 140 g/L cellobiose | 0 | 6.0 | 51 | |
| *Z. mobilis* ZM4 | 127 g/L | Glucose media | 300 g/L glucose | 10 g/L | – | 30 | (8, 40) |
| | 72 g/L | MR-MES medium | 150 g/L glucose | 5 g/L | – | 30 | (41) |
| | 43 g/L | ZMM | 100 g/L glucose | 0 | – | 30 | (42) |
| | 67 g/L | ZMM | 150 g/L glucose | 10 g/L | – | 37 | This study |
| | 8 g/L | ZMM | 150 g/L glucose | 0 | – | 37 | This study |
| | 45 g/L | ZMM-2 | 150 g/L glucose | 0 | – | 37 | This study |
| | 72 g/L | ZMM-2 | 160 g/L glucose | 0 | 6.0 | 37 | This study |

[a]"–" represents no pH control.

Similarly, engineered strains of *T. saccharolyticum* have been reported to produce 50–70 g/L ethanol in rich media such as TSC6 (4, 16). Using strain M1442, we observed ethanol titers up to 61 g/L when yeast extract was added to the defined MTC7 medium (Table 2). In the absence of yeast extract, the maximum titer dropped to 44 g/L (Table 2) in chemically defined growth medium (Table 2).

For *Z. mobilis*, included as a wild-type benchmark, we sought defined medium conditions that could support high ethanol titers without substrate limitation. Most prior studies of *Z. mobilis* in ZMM media have used relatively low glucose concentrations (<20 g/L) (41, 43). After confirming that *Z. mobilis* is auxotrophic for lysine and methionine (44), we grew it in ZMM-2 medium supplemented with these amino acids which improved ethanol production from 8 g/L to 45 g/L (Table 2). With active pH control, *Z. mobilis* was able to completely consume glucose up to ~160 g/L (File S1), but it was difficult to get the organism to reliably initiate growth at glucose concentrations above 160 g/L in ZMM-2 media. Here, "initiate growth'" refers to the ability of a microbe to reproducibly exit lag phase and enter exponential growth following inoculation. At glucose concentrations above ~160 g/L, growth initiation was inconsistent and often failed even after prolonged incubation (≥4 days), indicating impaired growth rather than delayed adaptation. Osmotic stress is a plausible contributing factor. Ultimately, we identified a condition where *Z. mobilis* achieved a final ethanol titer of 72 g/L when grown on 160 g/L glucose with 10 g/L initial ethanol added. The addition of a small initial amount of ethanol was necessary to observe cessation of fermentation with substrate still remaining.

It is interesting to note that yeast extract allows for higher titer ethanol production in all of these microbes, even without pH control (Table 2). Yeast extract is thought to help recover damaged cell membranes due to ethanol toxicity, allowing microbes to tolerate higher ethanol titers (45). The ability of yeast extract to support high-titer ethanol production may reflect supplementation with vitamins (e.g., thiamine) and amino acids, as well as, more generally, its effects on redox homeostasis and cofactor availability under stress conditions.

Despite these potential benefits, yeast extract is problematic for both applied and fundamental research. The expense of yeast extract often precludes its use in industrial fermentations, motivating the development of organisms that can achieve high titer in its absence. For fundamental research, several aspects of yeast extract make it difficult to work with. Yeast extract can exhibit variability between batches due to differences

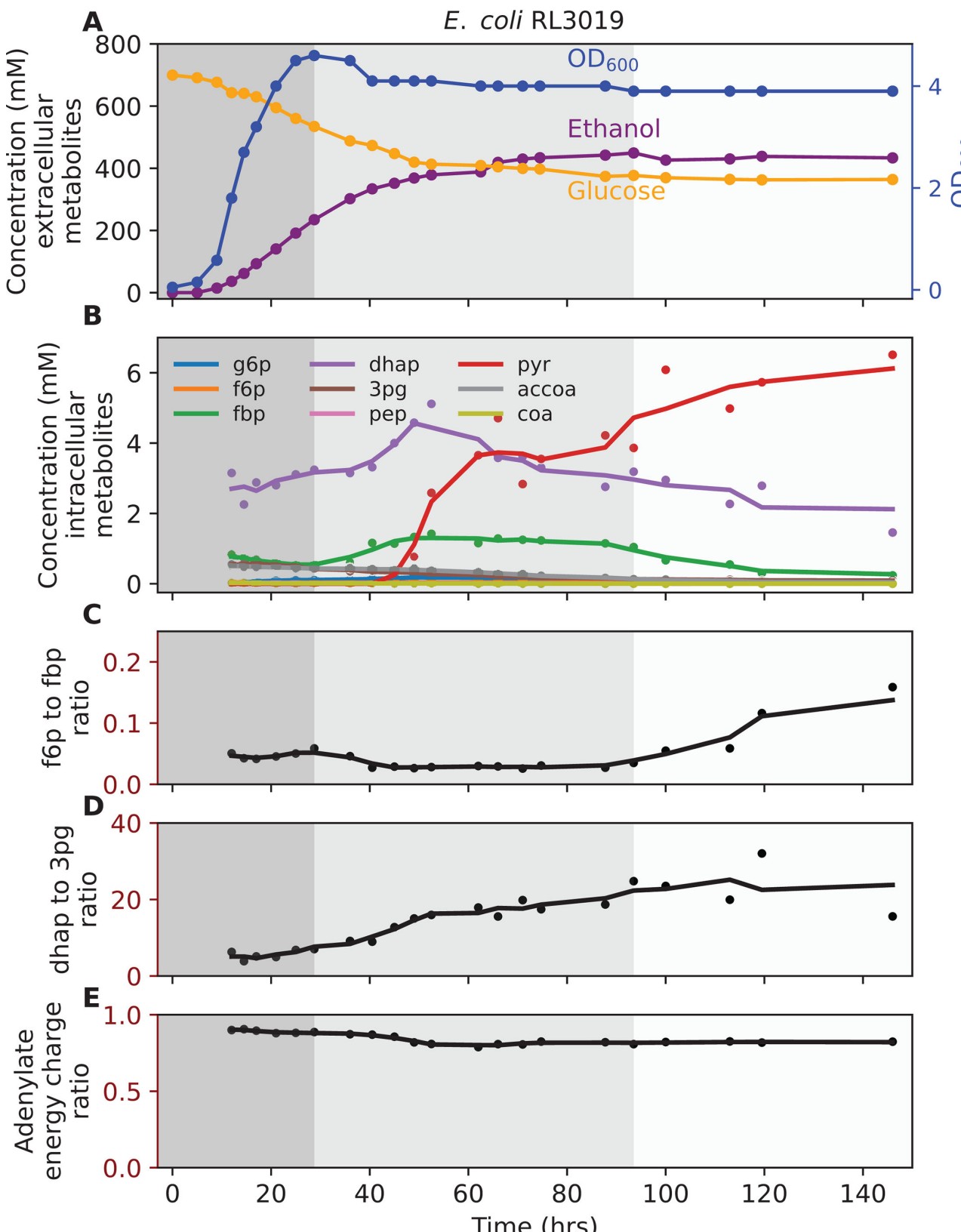

**FIG 1** Fermentation behavior of *E. coli* RL3019. (A) Concentration of extracellular metabolites. (B) Concentration of intracellular metabolites. (C) f6p to fbp ratio. (D) dhap to 3pg ratio. (E) Adenylate energy charge ratio. The fermentation was performed with 120 g/L glucose in M9 minimal medium at 37°C with pH maintained at 6.5 ± 0.05 by addition of 4 M potassium hydroxide. The shaded background represents different phases of fermentation: dark gray, growth coupled (Continued on next page)

**Fig 1 (Continued)**

fermentation; medium gray, growth uncoupled fermentation; and light gray, no-ethanol-production phases. One representative fermentation profile is shown ($n$ = 2). Figure S2 shows a biological duplicate of this experiment. File S1 (eco1_ext) shows additional extracellular metabolites data for this experiment. File S1 (eco1_int) shows intracellular metabolites concentration. In subplots B–E, each circle represents an individual measurement, and the line plot represents a trendline, which is smoothed using a rolling average with a window size of three data points. The abbreviations used are defined in "Enzymes and metabolites."

in raw material sources, production methods, and processing conditions (46). This lack of batch-to-batch consistency can lead to variations in its nutrient composition, including amino acids, peptides, and other bioactive compounds, which can cause batch effects that are difficult to control for. Furthermore, yeast extract interferes with LC-MS measurements of intracellular metabolites. Therefore, for our study, we proceeded using fermentation conditions with chemically defined media and with pH control. Although the final ethanol titers were lower than the maximum observed with complex media, these conditions allowed us to both observe the cessation of fermentation and to accurately quantify intracellular metabolites.

The replicates of data shown in Table 2 are provided in File S1 (fermentation_all_ext). The order of rows within the table reflects the approximate order in which experiments were performed, as we worked to identify suitable fermentation conditions.

## Fermentation behavior of *E. coli* RL3019, *T. saccharolyticum* M1442, and *Z. mobilis* ZM4

To study fermentation behavior under conditions where substrate availability does not limit ethanol production, we performed high-substrate batch fermentations: 120 g/L glucose for *E. coli*, 140 g/L cellobiose for *T. saccharolyticum*, and 160 g/L glucose with 10 g/L initial ethanol for *Z. mobilis*. For the purposes of analysis, we divided the fermentations into phases: growth-coupled fermentation, growth-uncoupled fermentation, and no ethanol production (Fig. 1A, 2A, and 3A), similar to what we have done before for *C. thermocellum* (17).

For *E. coli*, the final ethanol titer reached 448 ± 2 mM (20.6 ± 0.1 g/L), with 358 ± 8 mM (64.5 ± 1.4 g/L) glucose remaining at the end of fermentation (Fig. 1A). For *T. saccharolyticum*, the final ethanol titer reached 949 ± 6 mM (43.7 ± 0.3 g/L), with 203 ± 2 mM (36.5 ± 0.4 g/L) of glucose accumulating in the medium (Fig. 2A). Although cellobiose was fully consumed by ~90 h (Fig. 2A), fermentation was not substrate-limited because glucose, which is also a utilizable carbon source for *T. saccharolyticum* (47), remained available. For *Z. mobilis* fermentation, the final ethanol titer reached 1553 ± 15 mM (71.6 ± 0.7 g/L), with 174 ± 2 mM (31.3 ± 0.4 g/L) glucose still present at the end of fermentation (Fig. 3A).

Analysis of intracellular metabolites revealed that pyruvate accumulated progressively in *E. coli* and *T. saccharolyticum*, indicating inhibition of reactions downstream of pyruvate (Fig. 1B and 2B). To further probe glycolytic flux, we examined metabolite ratios that serve as indicators of bottlenecks between upper and lower glycolysis. In *E. coli*, the ratio of f6p to fbp increased during the late growth-uncoupled phase (Fig. 1C), consistent with reduced activity at the PFK step. In *T. saccharolyticum*, f6p was not detected, so the g6p to fbp ratio was used as a proxy and showed a similar upward trend (Fig. 2C). The dhap to 3pg ratio also increased as fermentation progressed in both microbes (Fig. 1D and 2D), consistent with restricted flux in lower glycolysis downstream of triose phosphate metabolism. In *T. saccharolyticum*, the pyruvate to acetyl-CoA ratio began increasing during the early–mid growth-uncoupled phase (Fig. 2D), suggesting reduced conversion of pyruvate to acetyl-CoA. Together, these data indicate that pyruvate accumulation occurs first, followed by the buildup of upstream intermediates extending to hexose phosphates. In *T. saccharolyticum*, the accoa to coa ratio decreased further (Fig. 2F), reinforcing the interpretation of a bottleneck at the pyruvate-to-acetyl-CoA step. Additionally, the ratios of oxidized to reduced nicotinamide cofactors (nad to nadh and nadp to nadph) decreased as fermentation progressed (Fig.

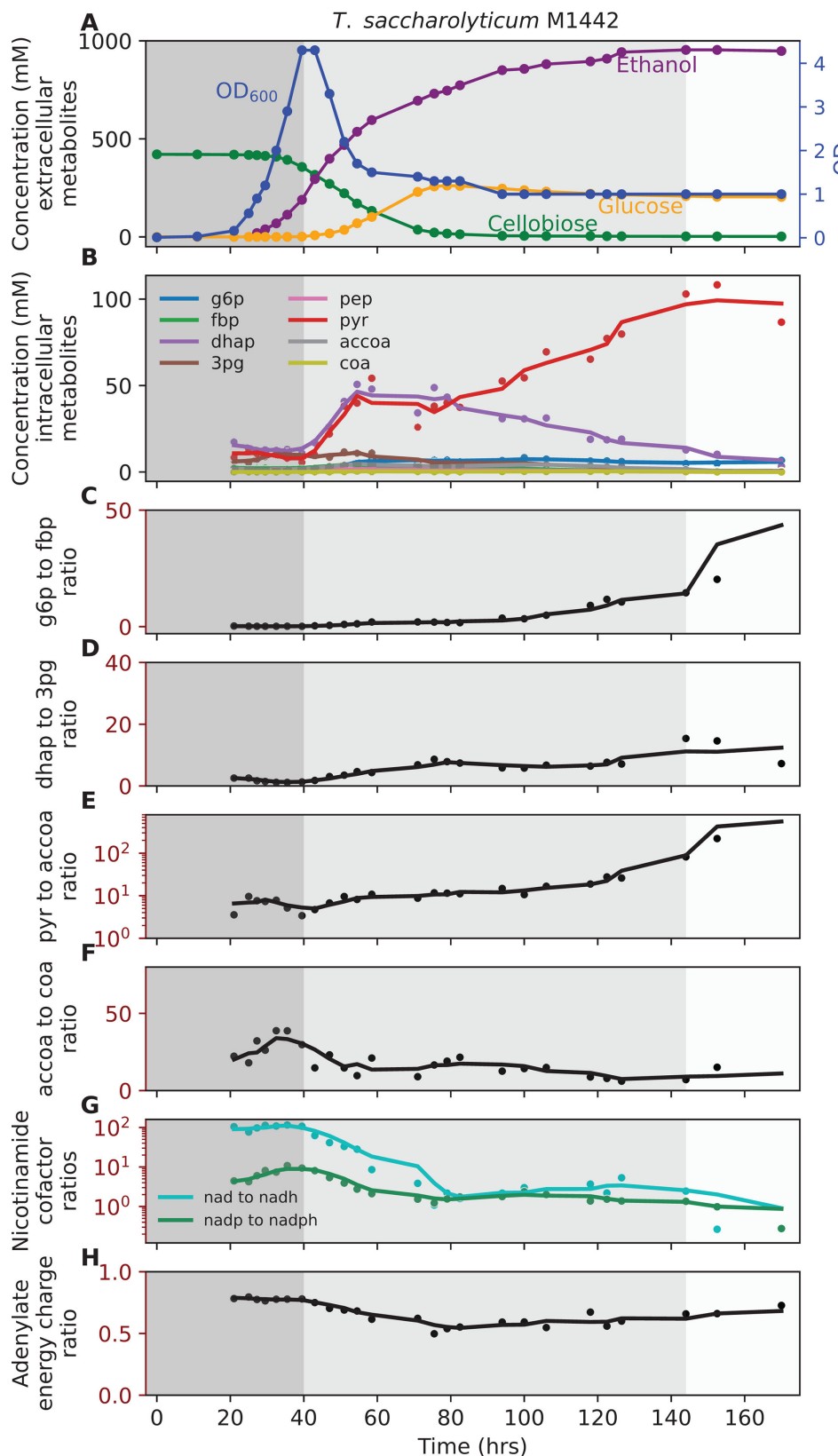

**FIG 2** Fermentation profile of *T. saccharolyticum* M1442. (A) Concentration of extracellular metabolites. (B) Concentration of intracellular metabolites. (C) g6p to fbp ratio. (D) dhap to 3pg ratio. (E) pyr to accoa ratio. (F) accoa to coa ratio. (G) Nicotinamide cofactor ratios. (H) Adenylate energy charge ratio. Fermentation was performed with 140 g/L cellobiose in

Fig 2 (Continued)

MTC-7 medium at 51°C, with pH maintained at 6.0 ± 0.05 by addition of 4 M potassium hydroxide. The shaded background represents different phases of fermentation: dark gray, growth-coupled fermentation; medium gray, growth-uncoupled fermentation; and light gray, no-ethanol-production phases. One representative fermentation profile is shown ($n = 2$). Figure S3 shows a biological duplicate of this experiment. File S1 (tsac1_ext) shows additional extracellular metabolites data for this experiment. File S1 (tsac1_int) shows intracellular metabolites concentration. In subplots B–H, each circle represents an individual measurement, and the line plot represents a trendline, which is smoothed using a rolling average with a window size of three data points. The abbreviations used are defined in "Enzymes and metabolites."

2G). By the end of fermentation, the average ratios were 2.5 for nad to nadh and 1.6 for nadp to nadph, values close to those reported previously for mid-log cells grown at lower substrate concentrations (2.1 and 1.4, respectively) (48). This further supports the conclusion that fermentation is inhibited at the pyruvate-to-ethanol pathway. In *E. coli* and *Z. mobilis*, metabolite ratios for the pyruvate to acetaldehyde step and nicotinamide cofactors could not be quantified because acetaldehyde and nadh were not detected by LC-MS.

By contrast, *Z. mobilis* did not accumulate pyruvate as fermentation progressed (Fig. 3B). The g6p to 6pgn ratio increased slightly after the growth-coupled phase (Fig. 3C), while other ratios showed transient increases followed by declines (Fig. 3D and E). The absence of sustained accumulation of intracellular metabolites suggests that the limitation of ethanol titer in this organism may not stem from central metabolic bottlenecks, but rather from other factors, such as inhibition of substrate uptake transporters. Across all three microbes, the adenylate energy charge showed a slight decrease during active ethanol production, but the physiological significance of these changes is not known (Fig. 1E, 2H, and 3F).

## Global overview of metabolite changes during growth and fermentation

In addition to specific intracellular metabolites with absolute quantification, our LC-MS platform gives us information about relative concentrations of dozens of other metabolites. By clustering metabolites based on relative abundance changes over time, we can observe patterns of metabolite changes (Fig. 4).

In *E. coli* RL3019, there is a relative lack of central carbon metabolites, suggesting that many of them are present at very low levels. In pyruvate metabolism, there is an initial accumulation of acetyl-CoA and asparagine, but as growth slows, a variety of pyruvate-derived amino acids start to accumulate. In nucleotide metabolism, there is a gradual transition from high-energy phosphates (ATP, GTP) to low-energy phosphates (AMP, GMP) to nucleotide breakdown products (uracil, inosine, hypoxanthine, xanthine, guanine) (Fig. 4). Note that these changes are emphasized by the row-wise normalization of the heatmap (Fig. 4), because they are scaled to the max and min values of the data. The heatmap emphasizes trends and changes. The absolute metabolite concentration measurements (Fig. 1 to 3) provide better comparison to known physiological states (e.g., adenylate charge) and between organisms.

In *T. saccharolyticum* M1442, glycolytic intermediates accumulate during the log phase. Immediately after growth stops, there is a large flux shift to the TCA cycle, and many glycolytic intermediates are depleted. As fermentation slows, there is a large accumulation of intermediates in the valine, leucine, and isoleucine biosynthesis pathway; however, there appears to be some kind of bottleneck, as the levels of the end-product amino acids are low. In energy cofactor metabolism, there is a spike in high-energy phosphates (ATP, GTP) right before growth stops and a spike of low-energy phosphates (AMP, GMP) right after growth stops. As fermentation slows and stops, there is a moderate accumulation of nucleotide degradation products (thymidine, uridine, uracil, and xanthosine) (Fig. 4).

In *Z. mobilis* ZM4, there is a long lag phase where intracellular metabolites are generally low. Similar to *T. saccharolyticum*, log-phase growth is associated with high

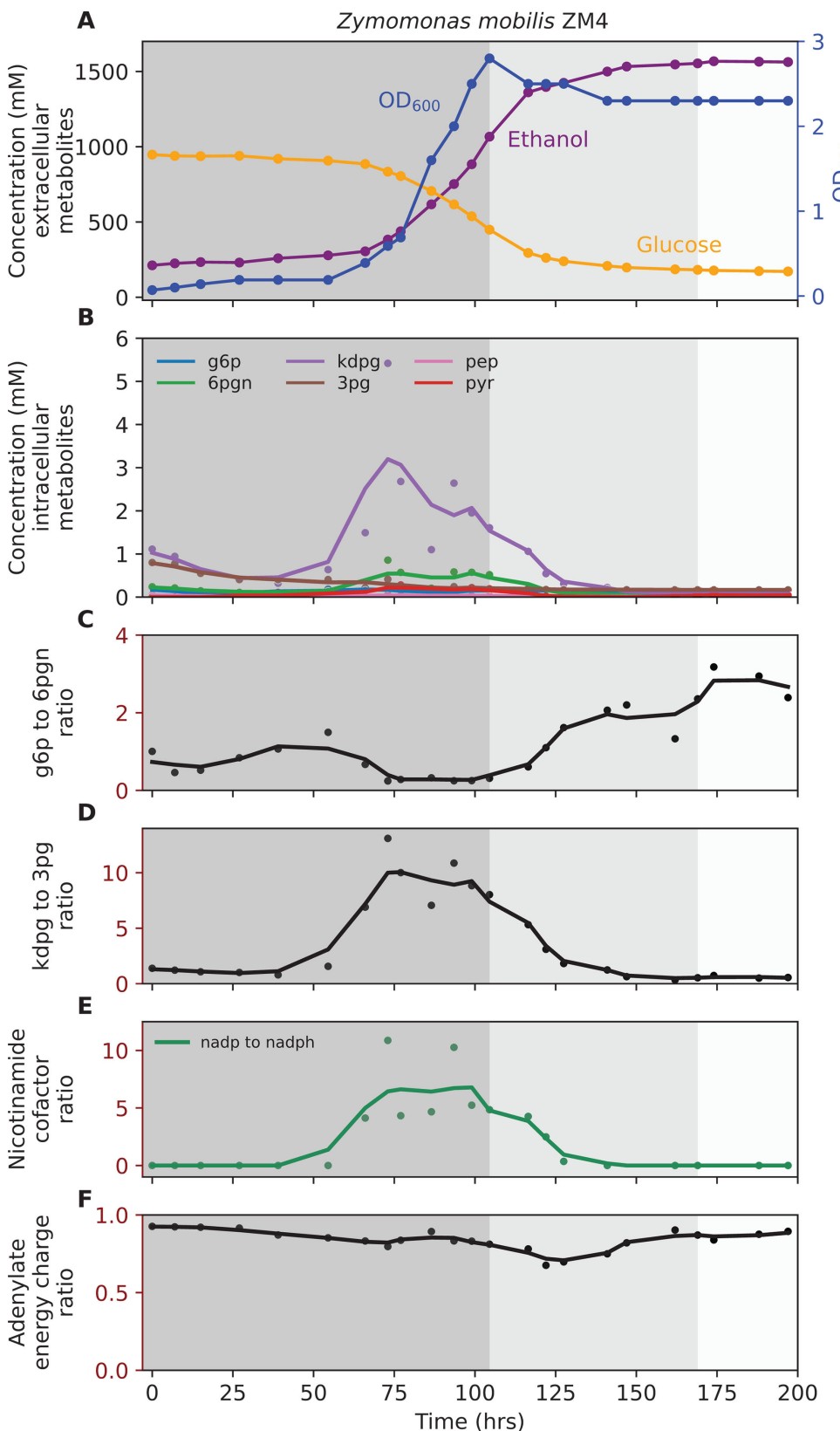

**FIG 3** Fermentation profile of *Z. mobilis* ZM4. (A) Concentration of extracellular metabolites. (B) Concentration of intracellular metabolites. (C) g6p to 6pgn ratio. (D) kdpg to 3pg ratio. (E) Nicotinamide cofactor ratio (F) Adenylate energy charge ratio Fermentation was performed with 160 g/L glucose in ZMM-2 medium at 37°C with pH maintained at 6.0 ± 0.05

Fig 3 (Continued)

by addition of 4 M potassium hydroxide. The shaded background represents different phases of fermentation: dark gray, growth coupled fermentation; medium gray, growth uncoupled fermentation; and light gray, no-ethanol-production phases. One representative fermentation profile is shown ($n$ = 2). Figure S4 shows a biological duplicate of this experiment. File S1 (zmm1_ext) shows additional extracellular metabolites data for this experiment. File S1 (zmm1_int) shows intracellular metabolites concentration. In subplots B–F, each circle represents an individual measurement, and the line plot represents a trendline, which is smoothed using a rolling average with a window size of three data points. The abbreviations used are defined in "Enzymes and metabolites."

levels of central carbon metabolites. After growth stops, upper glycolysis intermediates (fructose-6-phosphate, glucose-6-phosphate, and glucose-1-phosphate) accumulate, along with some pyruvate-derived metabolites (leucine/isoleucine, valine, and succinate). In energy cofactor metabolism, a similar pattern is observed compared to *T. saccharolyticum*, with a spike in high-energy phosphates (ATP, GTP) right before growth stops, a spike of low-energy phosphates (AMP, GMP) right after growth stops, and accumulation of nucleotide degradation products as fermentation stops (Fig. 4).

## Thermodynamic constraints highlight PYK as a limiting step in ethanologen *E. coli* and *T. saccharolyticum*, with *Z. mobilis* showing no clear bottleneck

In isolation, it can be difficult to interpret the physiological effect of changes in metabolite levels; therefore, we applied the max-min driving force (MDF) framework (13) to measured metabolite concentrations to understand how the systematic interaction of different metabolites can place thermodynamic equilibrium constraints on metabolism. We hypothesized that identifying thermodynamically constrained reactions would help identify potential mechanisms of product titer limitation.

The inputs to MDF analysis are the stoichiometric network and the concentrations of measured metabolites. From this, the Gibbs free energy of each reaction under standard conditions ($\Delta rG'^{\circ}$) is calculated using databases of standard thermodynamic potentials (29). Finally, the MDF score is calculated by optimizing concentrations of non-measured metabolites to maximize the minimum driving force. Essentially, this analysis asks: under the best-case scenario for non-measured metabolites (i.e., with respect to MDF), what reactions most constrain the thermodynamics of the overall pathway?

A pathway is considered thermodynamically feasible if its MDF score is positive, which also reflects the degree of kinetic constraint from backward flux. Positive MDF values indicate that all reactions can maintain sufficiently high driving forces to support forward flux, whereas MDF values near or below zero suggest that at least one reaction is at thermodynamic equilibrium or thermodynamically infeasible (13). Importantly, MDF analysis identifies where reactions first become thermodynamically infeasible but does not, by itself, distinguish among the biological mechanisms responsible for the loss of flux. Because loss of flux for any underlying reason is expected to cause reversible reactions to rapidly approach equilibrium, the observation of local thermodynamic equilibrium is most appropriately interpreted as a diagnostic reflection of flux cessation. At each fermentation time point, MDF scores were calculated using the eQuilibrator Python library (29) (https://gitlab.com/equilibrator/equilibrator-pathway). The reactions were set up as shown in Fig. 5, and the reaction stoichiometries are provided in File S2.

MDF analysis yields two outputs. The first is the overall MDF score for the pathway, which should be positive whenever forward flux occurs. The second is the identification of bottleneck reactions (highlighted in red in Fig. 6), representing the reactions that are determining the MDF value (i.e., the most thermodynamically limiting steps) under the measured metabolite concentrations. As expected, during the growth-coupled and growth-uncoupled fermentation phases, MDF scores were largely positive in *E. coli*, aside from a few early points showing high variability due to low cell density. In *Z. mobilis*, MDF scores decreased slightly as fermentation progressed but remained positive throughout, including during the no-ethanol-production phase. In contrast, *T. saccharolyticum*

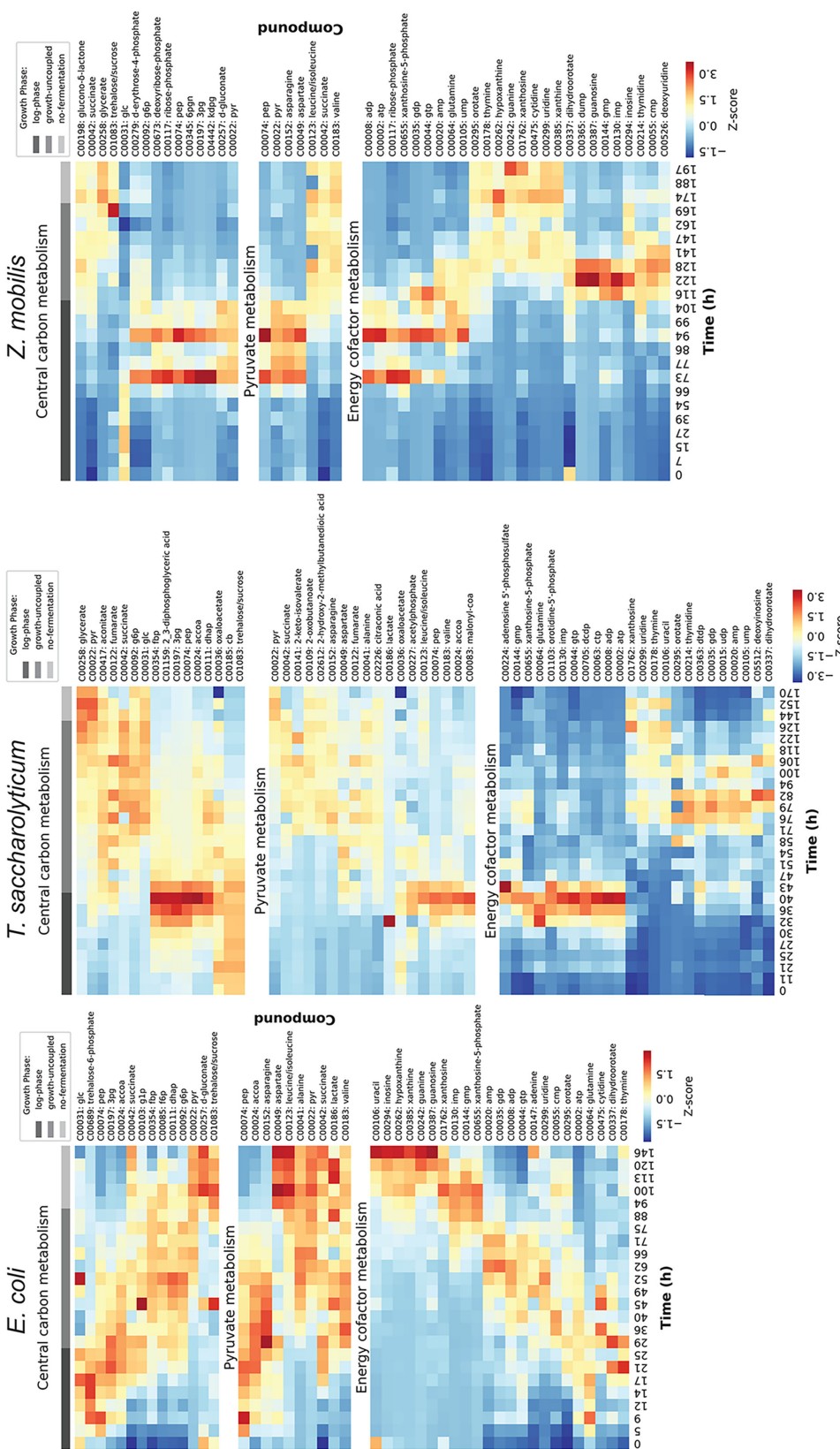

**FIG 4** Relative abundance of metabolites during fermentation. Untargeted metabolite analysis was performed for all samples. Metabolites were identified by *m/z* ratio and retention time. For each metabolite, peak area values were normalized by calculating z-scores across all time points, allowing comparison of temporal dynamics independent of absolute abundance

Fig 4 (Continued)

levels. Metabolites were hierarchically clustered using average linkage clustering with correlation distance as the similarity metric. Red shading indicates high abundance. Blue shading indicates low abundance. The growth phase is indicated by shaded bars at the top of the columns: dark gray corresponds to growth-coupled fermentation, medium gray corresponds to growth-uncoupled fermentation, and light gray corresponds to no fermentation. Only one replicate is shown here. Both replicates and all metabolites are shown in supporting Fig. S5. Underlying data are available as File S3.

exhibited several intervals of negative MDF scores despite active substrate conversion to ethanol, suggesting these values likely reflect experimental error and prompting additional analyses to identify possible sources of this discrepancy.

## Error analysis of MDF scores in *E. coli* and *T. saccharolyticum*

We considered two sources of error for our MDF measurements of negative scores in engineered strains of *E. coli* and *T. saccharolyticum*: (i) random measurement error (equally distributed across all metabolites) or (ii) specific errors for a single metabolite.

For the first approach, we investigated whether simultaneously relaxing the concentrations of all quantified metabolites by applying multiplicative factors to the concentration bounds would alter the MDF scores. The lower bound for each metabolite was set to the measured value divided by the chosen factor, while the upper bound was set to the measured value multiplied by the same factor. For *T. saccharolyticum*, MDF scores remained negative at the measured concentrations but became positive once the bounds were expanded beyond a 1.5-fold change (Fig. S7). At later stages of fermentation, larger relaxations were required, and progressively increasing the bounds (up to 100-fold) raised the MDF values but did not change the qualitative trend of declining thermodynamic feasibility after ethanol production ceased (Fig. S7). In the case of *E. coli*, MDF scores during the later phase of fermentation became positive with as little as a 2-fold relaxation. With more than a 5-fold change in metabolite concentrations, even the late fermentation points were rendered thermodynamically feasible. Since only large deviations from measured concentrations were sufficient to reverse the negative MDF scores, we next examined the effect of measurement errors in single metabolites.

For the second approach, we investigated whether relaxing the concentration bounds of individual measured metabolites would affect the MDF scores. For each metabolite, the lower bound was set to 1 µM and the upper bound to 100 mM. As shown for *E. coli* (Fig. 7), relaxing the bounds of pep, pyr, atp, or adp individually was sufficient to raise the MDF score to higher values, even during the no-ethanol-production phase. These metabolites are all associated with the PYK reaction, and the observation in this relaxation experiment is consistent with our identification of the PYK reaction being in local thermodynamic equilibrium (Fig. 6, 146-hour time point sub-panel).

In the case of *T. saccharolyticum*, we can assess the effect on metabolite-bound relaxation on the two negative-MDF (i.e., thermodynamically infeasible) regions separately (Fig. 8).

i. The reaction infeasibility observed between 29 and 57 h could be resolved by changing the bounds for etoh, accoa, coa, nad, nadh, nadp, and nadph. As mentioned above (at 43 h; Fig. 8 for *T. saccharolyticum*), the metabolic bottleneck is seen at the reactions downstream of pyruvate, and changing the bounds of any of these metabolites alleviates the issue.
ii. The reaction infeasibility after ethanol production stops. Although we could not resolve the negative or lower MDF values for all time points as we did for *E. coli*, we were able to improve the feasibility at the first time point by relaxing constraints on the bounds for pep, pyr, atp, and adp. This improvement is also attributed to the fact that these metabolites are involved in the PYK reaction and changing any of them would increase the feasibility of this reaction.

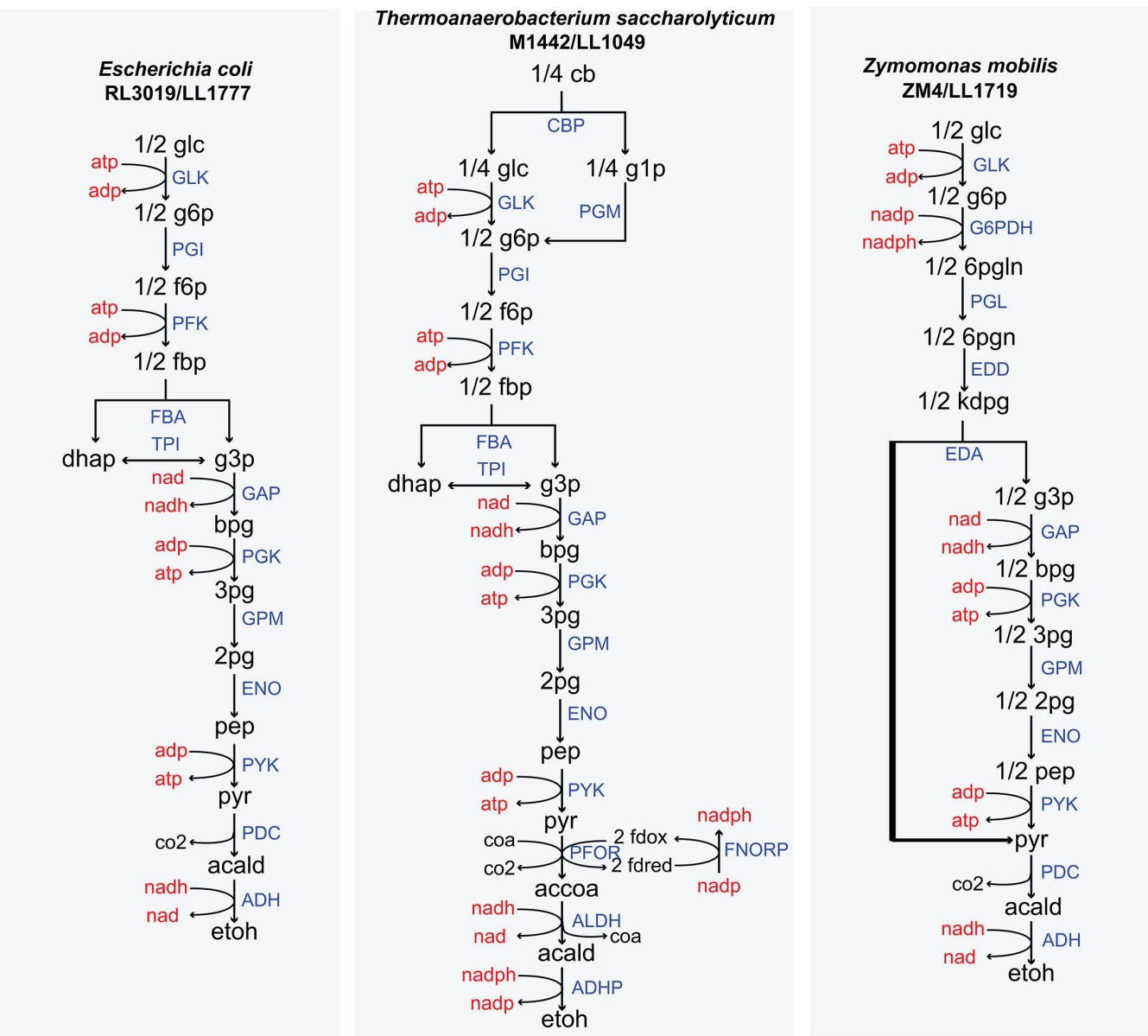

**FIG 5** Graphical representation of *E. coli* RL3019, *T. saccharolyticum* M1442, and *Z. mobilis* ZM4 fermentation pathway. Note that *T. saccharolyticum* does not have the FNORP reaction. However, the reactions involved in that specific location have stoichiometry similar to canonical FNORP (49). Blue capitalized text represents reactions, red text indicates cofactors, and black text represents metabolites. The abbreviations used are defined in "Enzymes and metabolites."

## DISCUSSION

The goal of this study was to understand factors that limit ethanol titer in microbial fermentations. A key limitation in much prior work on this topic is the lack of high-quality intracellular metabolite data collected during fermentations in which ethanol was produced at high titer. By combining both high-titer fermentations and quantitative intracellular metabolite quantification, the data collected here therefore represents a unique window into the metabolic changes that occur as fermentation slows and eventually stops. Furthermore, by observing this phenomenon in several different species (*E. coli*, *T. saccharolyticum*, and *Z. mobilis*), we can get hints about the generalizability of our insights across different species.

This topic has also been extensively studied in yeast (*Saccharomyces cerevisiae*). In the brewing literature, the phenomenon of fermentation slowing or stopping before

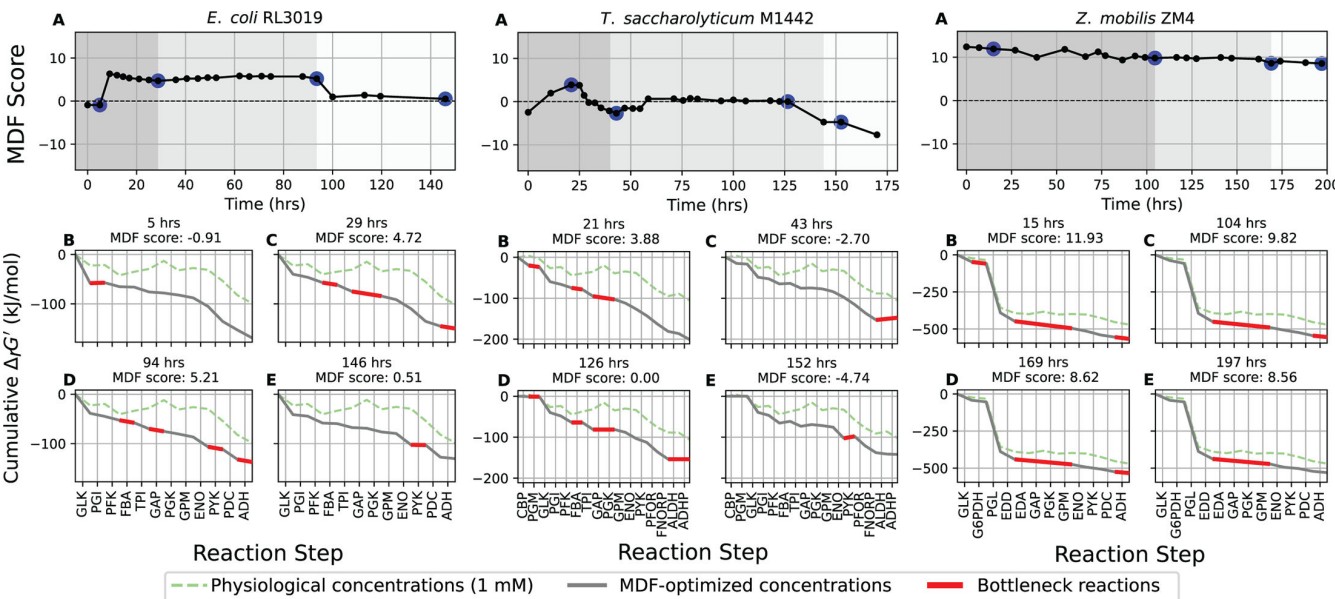

**FIG 6** Max-min driving force (MDF) scores during fermentations of *E. coli* RL3019 (left), *T. saccharolyticum* M1442 (middle), and *Z. mobilis* ZM4 (right). MDF scores were calculated at each time point during the course of fermentation. The blue circles represent the specific time points for which MDF scores are plotted. In the top row of panel A, the shaded backgrounds indicate different phases of fermentation: dark gray, growth-coupled fermentation; medium gray, growth-uncoupled fermentation; and light gray, no ethanol production. For each organism, the four lower subplots (B, C, D, and E) represent cumulative ΔrG' for the blue-circle time points in the MDF score chart. The green dotted line shows cumulative ΔrG' when metabolite concentrations are fixed at 1 mM, the gray line shows values based on measured concentrations, and the red line highlights the bottleneck reactions responsible for changes in MDF scores. Biological replicates of these experiments are shown in Fig. S6. The abbreviations used are defined in "Enzymes and metabolites."

the substrate is consumed is known as a "stuck" fermentation (50, 51). Several factors have been proposed as causes of stuck fermentations, including deficiencies in nitrogen,

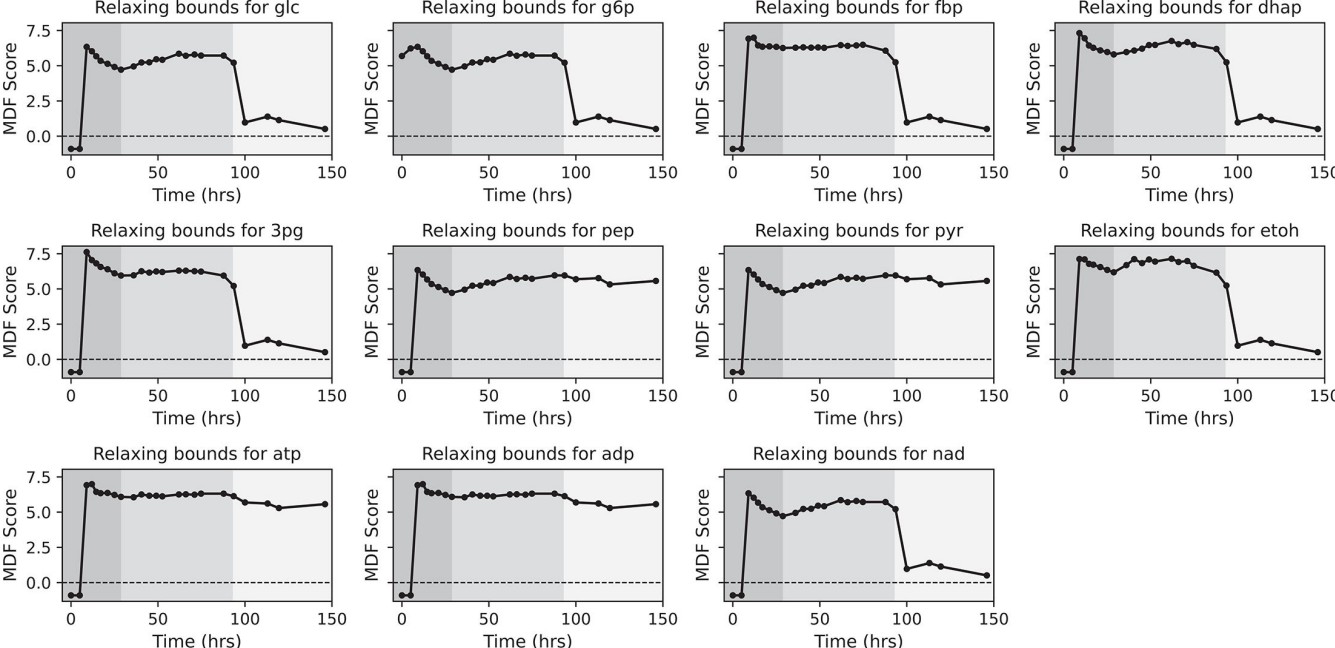

**FIG 7** Max-min driving force (MDF) score by relaxing the bounds of measured metabolites during *E. coli* RL3019 fermentation. The shaded background represents different phases of fermentation: dark gray, growth-coupled fermentation; medium gray, growth-uncoupled fermentation; and light gray, no-ethanol-production phases. The abbreviations used are defined in "Enzymes and metabolites."

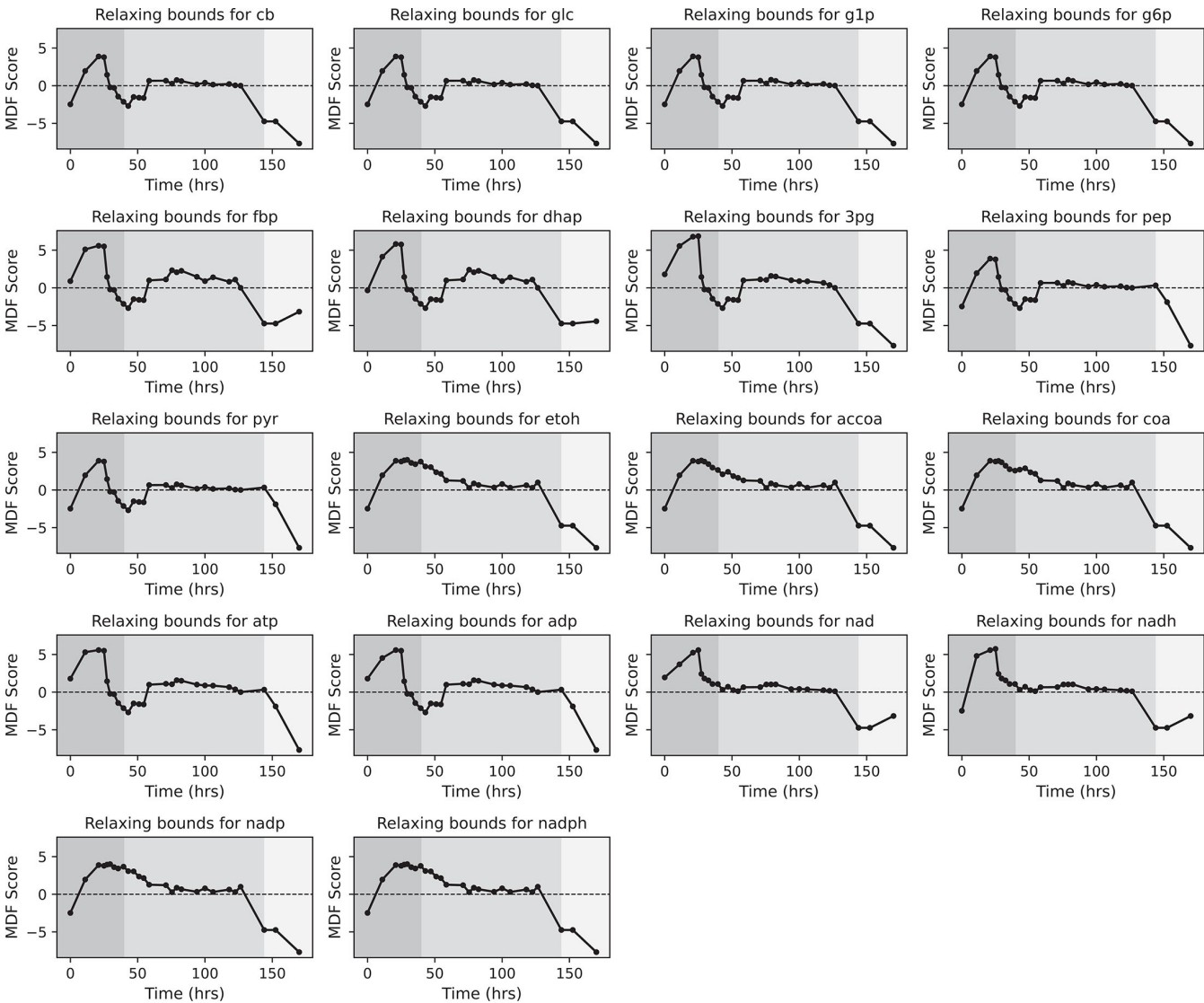

**FIG 8** Max-min driving force (MDF) score by relaxing the bounds of measured metabolites during *T. saccharolyticum* M1442 fermentation. The shaded background represents different phases of fermentation: dark gray, growth-coupled fermentation; medium gray, growth-uncoupled fermentation; and light gray, no-ethanol-production phases. The abbreviations used are defined in "Enzymes and metabolites."

oxygen, vitamins and minerals, ethanol inhibition, and membrane damage (50); however, the mechanisms by which these factors affect metabolism are not well understood. It has been shown that ethanol inhibits key glycolytic enzymes, such as phosphofructokinase, phosphoglycerate kinase, and pyruvate decarboxylase, leading to a gradual slowdown in fermentation (52), but the extent to which this is responsible for the stuck fermentation phenotype is not known. One of the more well-supported mechanisms explaining stuck fermentations is a lack of substrate transport. Sugar transporters in yeast are known to have a low half-life, requiring rapid biosynthesis to maintain transport capacity. Thus, a cessation of protein synthesis can rapidly eliminate substrate transport (51).

Previously, cessation of ethanol production in *Z. mobilis* was suggested to result from enzyme inhibition, membrane disruptions, and metabolic imbalances (45, 52). Although all of these factors may be present at high enough ethanol concentrations, we are most interested in identifying the factor(s) that first cause fermentation to stop. Our observation of a relative lack of accumulation of all intracellular metabolites in *Z. mobilis* when fermentation stops suggests that inhibition of individual enzymes or imbalances of

metabolite ratios is unlikely to cause this phenotype (Fig. 3). Instead, we propose that a lack of substrate transport is the primary cause. However, we cannot completely rule out the possibility of metabolite leakage playing a role.

Previous studies on ethanologenic strains of *E. coli* have suggested that the *Z. mobilis* Pdc enzyme is inhibited by ethanol (52, 53), resulting in accumulation of pyruvate at high ethanol titers and eventually the cessation of fermentation. Our observations in *E. coli* are largely consistent with prior studies and suggest that loss of flux through the PDC reaction is a plausible initiating event responsible for the stopping phenotype (Fig. 1). Interestingly, however, *Z. mobilis* itself does not display strong inhibition of the pyruvate-to-ethanol pathway under our tested conditions, as no pyruvate accumulation was observed during fermentation (Fig. 3). This is particularly puzzling, since the heterologous ethanol production pathway in *E. coli* RL3019 consists of the *pdc* and *adh* genes from *Z. mobilis*. One possible explanation may be related to cofactor biosynthesis. The Pdc enzyme requires thiamine pyrophosphate, and its native biosynthesis pathways may be differently inhibited by ethanol in *E. coli* vs. *Z. mobilis*. This could also explain the stimulatory effect of yeast extract (which contains high levels of thiamine) on ethanol titer (Table 2). Another possibility is that the Pdc enzyme is affected by pH (54), and differences in intracellular pH between *E. coli* and *Z. mobilis* may affect activity. Understanding the reason for this observed difference between *E. coli* and *Z. mobilis* is an important area for future investigation. Furthermore, the direct comparison of *E. coli* and *Z. mobilis* provides an indication of the extent to which the industrially relevant ethanol production phenotype has been successfully transferred. At high substrate concentrations in chemically defined media, *Z. mobilis* exhibits more than a 3-fold higher ethanol titer compared to *E. coli* (Table 2). Therefore, it appears that simply transferring the pyruvate-to-ethanol pathway (*pdc* and *adhB* genes) is only sufficient to partly transfer the ethanol-producing ability of *Z. mobilis* (52, 53).

Compared to either *E. coli* or *Z. mobilis*, the metabolism of *T. saccharolyticum* is relatively poorly understood. The wild-type organism is known to be relatively ethanol-sensitive. In a comparison among six other bacteria, it was the most ethanol-sensitive and the only one unable to initiate growth at ethanol concentrations of 25 g/L or higher (55). Interestingly, it appears to have a latent ethanol production pathway that allows for both high yield and titer production. This pathway can be activated by disrupting enzymes associated with acetate and lactate production (56, 57). Subsequently, mutations are observed in the *hfs* gene cluster that appear to increase ethanol yield by increasing electron transfer from ferredoxin to NAD(P)$^+$ (58, 59). Additional mutations in the *adhE* gene disrupt the activity of the ADH domain of AdhE (60), forcing it to rely on the NADPH-linked AdhA enzyme instead (61) and effectively switching the cofactor specificity of the ADH reaction from NADH to NADPH.

Previously, we found that in wild-type *T. saccharolyticum*, adding ethanol during fermentation causes a substantial increase in the dhap to 3pg ratio, indicative of a metabolic bottleneck at the GAPDH reaction (62); however, that effect was decreased in this work, suggesting that the mutations involved in the development of the ethanologen strain tested here (strain M1442) substantially reduced or eliminated that bottleneck (Fig. 2). In fact, this outcome is consistent with our previous thermodynamic analysis of the effects of changing the cofactor specificity of the ADH reaction (63).

For the ethanologenic strain of *T. saccharolyticum* (strain M1442), no prior studies have examined changes in its metabolism during high-titer fermentation. Here, we observe an increase in pyruvate associated with the cessation of fermentation. The most likely explanation is that fermentation stopped due to loss of function in the pyruvate-to-ethanol pathway, similar to what was observed in *E. coli*. The relatively high increase in the pyruvate to acetyl-CoA ratio, compared to the relatively steady ratio of acetyl-CoA to CoA, suggests that a lack of flux through the PFOR reaction may be the immediate cause of fermentation cessation. Previously, we have shown that Pfor enzymes from different organisms differ in their ability to enable high-titer ethanol production (16). However, the PforA enzyme from *T. saccharolyticum* enabled the highest ethanol titers. It is also

possible that the lack of flux through the PFOR reaction is due to inhibition of reactions necessary for electron transfer from ferredoxin to NAD(P)$^+$, including NfnAB, HfsD, or HydA (59).

While the observed metabolite profiles and MDF signatures localize where thermodynamic infeasibility first emerges, they do not establish causality with respect to the underlying biological mechanism. Enzyme deactivation, redox imbalance, and limitations in substrate transport are non-mutually exclusive hypotheses that could give rise to the observed loss of flux and associated thermodynamic signatures.

The role of energy charge in cessation of fermentation is still ambiguous. For all three organisms we studied, the adenylate energy charge was relatively constant throughout fermentation in absolute terms (Fig. 1 to 3). On the other hand, the heatmap data show trends in energy cofactors associated with changes in growth phase. Although it is possible that relatively small changes in adenylate charge could have a large effect on the physiological response of the cell (Fig. 4), our current data do not allow us to draw strong conclusions one way or the other.

A critical factor that we identified via our error analysis of our MDF calculations was the importance of accurate cofactor quantification, particularly the nicotinamide cofactors (NADH, NAD$^+$, NADPH, and NADP$^+$). The large number of reactions in which these metabolites participate often gives them a large effect on MDF calculations. Such errors may create the appearance of pathway infeasibility even when flux is clearly occurring. Thus, rigorous metabolite sampling and analytical precision are essential when linking intracellular metabolite levels to thermodynamic constraints. Furthermore, these cofactors can also be rapidly interconverted during the metabolite quenching process. Since there are practical limits to the speed at which metabolite quenching can be performed, future experiments may benefit from the inclusion of orthogonal methods of nicotinamide cofactor quantification, such as redox-active sensors (64, 65) or dedicated redox reporter enzymes (66, 67).

Looking at all three strains together, several consistent patterns emerge. During rapid growth, relative concentrations of central metabolites are high. These pools start to deplete almost immediately after growth stops (Fig. 4; Fig. S5), potentially due to a lack of substrate transport. Thus, one of the most profound shifts in metabolism during batch fermentations is the shift from growth to non-growth. Once the cells have shifted to non-growth fermentation, the rate of product formation gradually slows down until it eventually stops completely. Unlike with the growth/no-growth transition, the metabolic signature of this transition is less abrupt (except perhaps for *E. coli*). Fermentation appears to stop due to a lack of flux through the pyruvate-to-ethanol pathways. In *E. coli* and *T. saccharolyticum*, this pyruvate accumulation then leads to a local thermodynamic equilibrium at the upstream (PYK) reaction, and this blockage eventually propagates upstream to upper glycolysis. In all three organisms, the final stage of cessation of fermentation is associated with an accumulation of nucleotide degradation products. The reasons for this are not clear, but one potential explanation is a loss of proton motive force, leading to acidification of the cytoplasm.

The key benefit of the thermodynamic analysis is that it allows us to build a causal chain between the observed changes in metabolite levels and a fundamental physical constraint (i.e., local thermodynamic equilibrium) that could explain the cessation of fermentation. The use of intracellular metabolite quantification under high-titer production conditions, combined with thermodynamic analysis, is a promising technique to provide insights into metabolic mechanisms of product titer limitation. Important areas for future research include more accurate quantification of energy and redox cofactors (ATP/ADP/AMP, GTP/GDP/GMP, NAD(P)$^+$/NAD(P)H), development of more robust pyruvate-to-ethanol production pathways, and understanding the remaining factors that allow *Z. mobilis* to produce higher ethanol titers compared to *E. coli*. Direct measurements of enzyme activity, substrate transport capacity, and intracellular redox dynamics would be required to distinguish among these mechanisms and represent

important directions for future work enabled by the diagnostic framework developed here.

## ACKNOWLEDGMENTS

This work was supported by the Center for Bioenergy Innovation (CBI), U.S. Department of Energy, Office of Science, Biological and Environmental Research Program under Award Number ERKP886.

B.D.S.: Conceptualization; Data curation; Formal analysis; Investigation; Methodology; Validation; Visualization; Writing-original draft. E.T.: Data acquisition and curation; Investigation. D.M.S.: Data acquisition. D.A.-N.: Data curation; Review; and Editing. L.R.L.: Funding acquisition; Project administration; Writing-review and editing. D.G.O.: Writing-review and editing; Supervision; Project administration; Funding acquisition.

## AUTHOR AFFILIATIONS

[1]Thayer School of Engineering, Dartmouth College, Hanover, New Hampshire, USA

[2]Center for Bioenergy Innovation, Oak Ridge National Laboratory, Oak Ridge, Tennessee, USA

[3]Department of Bacteriology, University of Wisconsin-Madison, Madison, Wisconsin, USA

[4]Great Lakes Bioenergy Research Center, University of Wisconsin-Madison, Madison, Wisconsin, USA

[5]Terragia Biofuels Inc., Hanover, New Hampshire, USA

## AUTHOR ORCIDs

Bishal Dev Sharma http://orcid.org/0000-0002-8326-4444
Eashant Thusoo http://orcid.org/0009-0001-0088-4235
Daniel Amador-Noguez http://orcid.org/0000-0002-3568-3070
Daniel G. Olson http://orcid.org/0000-0001-5393-6302

## FUNDING

| Funder | Grant(s) | Author(s) |
| --- | --- | --- |
| Biological and Environmental Research | ERKP886 | Bishal Dev Sharma |
| | | Eashant Thusoo |
| | | David M. Stevenson |
| | | Daniel Amador-Noguez |
| | | Lee R. Lynd |
| | | Daniel G. Olson |

## AUTHOR CONTRIBUTIONS

Bishal Dev Sharma, Conceptualization, Data curation, Formal analysis, Investigation, Methodology, Validation, Visualization, Writing – original draft, Writing – review and editing | Eashant Thusoo, Data curation, Investigation | David M. Stevenson, Investigation | Daniel Amador-Noguez, Data curation, Writing – review and editing | Lee R. Lynd, Funding acquisition, Project administration, Writing – review and editing | Daniel G. Olson, Conceptualization, Data curation, Funding acquisition, Project administration, Supervision, Visualization, Writing – review and editing

## ADDITIONAL FILES

The following material is available online.

## Supplemental Material

**File S1 (mSystems00074-26-s0001.xlsx).** Metabolite concentrations for *E. coli*, *T. saccharolyticum*, and *Z. mobilis* during the course of fermentation.

**File S2 (mSystems00074-26-s0002.xlsx).** Stoichiometry of reactions used for MDF analysis.

**File S3 (mSystems00074-26-s0003.xlsx).** Raw and quantified LCMS data.

**File S4 (mSystems00074-26-s0004.xlsx).** KEGG metabolite groups.

**File S5 (mSystems00074-26-s0005.txt).** Figure generation code: Jupyter notebook ipynb file.

**Supplemental material (mSystems00074-26-s0006.docx).** Figures S1-S7 and descriptions of Files S1-S5.

## Open Peer Review

**PEER REVIEW HISTORY (review-history.pdf).** An accounting of the reviewer comments and feedback.

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
