## [Reviewer comments · mSystems]

Metabolic imbalance limits fermentation in microbes engineered for high-titer ethanol production

Bishal Sharma, Eashant Thusoo, David Stevenson, Daniel Amador-Noguez, Lee Lynd, and Daniel Olson

Corresponding Author(s): Daniel Olson, Dartmouth College

Review Timeline:

Submission Date:

January 20, 2026

Accepted:

February 7, 2026

Editor: Pablo Ivan Nikel

Reviewer(s): Disclosure of reviewer identity is with reference to reviewer comments included in decision letter(s). The following individuals involved in review of your submission have agreed to reveal their identity: Uldis Kalnenieks (Reviewer #1)

Transaction Report:

DOI: <https://doi.org/10.1128/msystems.00074-26>

Re: mSystems00074-26 (**Metabolic imbalance limits fermentation in microbes engineered for high-titer ethanol production**)

Dear Dr. Olson:
Dear Daniel:

I am delighted to report that I am accepting your manuscript for publication in mSystems. Your paper will first be checked to make sure all elements meet the technical requirements. ASM staff will contact you if anything needs to be revised before copyediting and production can begin. Otherwise, you will be notified when your proofs are ready to be viewed.

Cover Image Submissions: If you would like to submit a potential Cover Image, please email a file and a short legend to mSystems@asmusa.org. Please note that we can only consider images that (i) the authors created or own and (ii) have not been previously published. By submitting, you agree that the image can be used under the same terms as the published article. Image File requirements: TIF/EPS, 7.5 inches wide by 8.25 inches tall (at least 2,250 pixels wide by 2,475 pixels tall), minimum 300 dpi resolution (600 dpi preferred), RGB, and no figure elements, e.g., arrows or panel labels. The legend should be a short description of the image, 1-2 sentences recommended. Please download and use this interactive template in Adobe to ensure that your proposed cover image meets our size requirements (<https://journals.asm.org/pb-assets/pdf-text-excel-files/ASM-Interactive-Sizing-Cover-Template-1715689791.pdf>).

Thank you for submitting your paper to mSystems!

Sincerely, Pablo

Prof. Dr. Pablo Ivan Nikel
Editor
mSystems

Reviewer #1 (Comments for the Author):

The authors have properly addressed my objections.

Reviewer #2 (Comments for the Author):

The authors have addressed all of my previous comments.